# TAMPERING DETECTION IN PRE-TRAINED ENCODERS VIA FINGERPRINT TWINS

## ABSTRACT

Encoder-as-a-Service (EaaS) enables pre-trained encoders to be shared across tasks, reducing cost but introducing integrity risks when models are modified without notice. Detecting tampering is difficult under a strict black-box setting, where the encoder is hidden within unknown pipelines and only application outputs are observable. Existing fingerprinting methods fail under these conditions as they require model predictions or task-specific information.

We present a novel fingerprinting framework for black-box encoder verification, *grounded in a theoretical insight that larger embedding divergence increases the likelihood of downstream output differences*. Building on this principle, we construct *fingerprint twins*—paired inputs that produce nearly identical embeddings on an intact encoder but diverge sharply after tampering. We simulate realistic changes using *importance-aware perturbations* and optimize twins to maximize KL divergence while constraining perturbations within an $\epsilon$-ball for natural appearance. Experiments across datasets and encoder types demonstrate reliable, task-agnostic detection with negligible impact on utility.

## 1 INTRODUCTION

Encoder-as-a-Service (EaaS) enables pre-trained encoders—whether vision, language, or multimodal—to be reused across diverse tasks such as classification, retrieval, and alignment (Radford et al., 2021), significantly reducing development costs. However, this shared deployment model introduces integrity risks: an encoder may be modified without authorization, either maliciously (e.g., injecting a backdoor or embedding hidden tracking) or benignly (e.g., replacing with a cheaper compressed model) (Liu et al., 2018; Gu et al., 2019; Rokh et al., 2023). Such modifications often preserve normal performance, making detection difficult. Unlike end-to-end models—whose predictions are observable—encoders operate upstream, hidden behind proprietary pipelines, rendering their behavior opaque. Verification must therefore rely solely on the application's outputs under a strict black-box setting, where internal components and downstream logic remain unknown. This raises a fundamental question: *can we determine whether an embedded encoder has been modified using only black-box access to application outputs?*

Existing fingerprinting techniques for tamper detection, though effective for end-to-end classifiers, do not fit this setting. They rely on observing model predictions on fingerprint samples (Aramoon et al., 2021; Bai et al., 2024; Wang et al., 2022; He et al., 2019). In our scenario, verification cannot access the encoder's own predictions—only the application's final outputs after unknown downstream processing. Moreover, downstream components are typically unavailable when the encoder owner generates fingerprints, making task-specific fingerprint design infeasible. Consequently, black-box verification of encoder integrity across diverse and unseen applications remains an open challenge.

We address this challenge with a novel fingerprinting framework for black-box integrity verification of pre-trained encoders. The core idea is to construct *fingerprint twins*—pairs of fingerprints that produce highly similar representations on an intact encoder but diverge significantly when the encoder is modified (Fig. 1). This divergence increases the likelihood of observable differences in final application outputs, even though those outputs are generated through unknown and potentially complex downstream pipelines. The rationale is grounded in a fundamental property: if any downstream function is Lipschitz-continuous with respect to encoder embeddings, larger embedding divergence

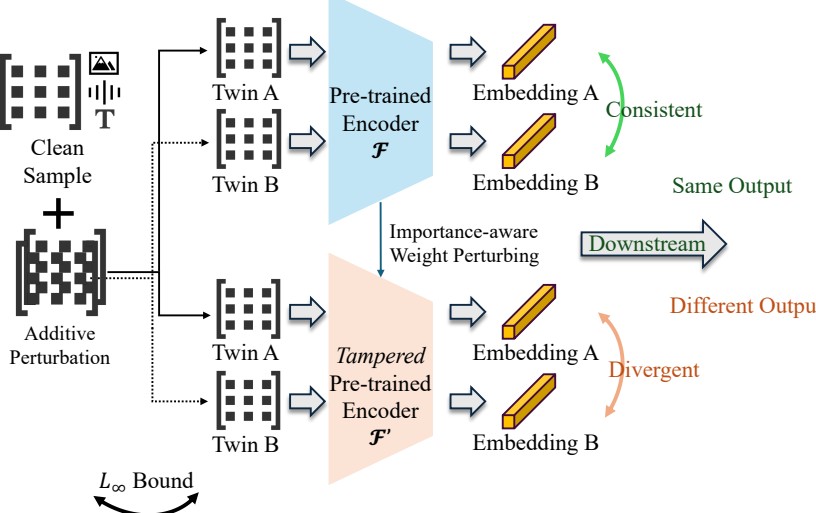

Figure 1: Overview of fingerprint twins. Sample pairs are crafted to produce similar embeddings on an intact encoder but diverge sharply under tampering, with importance-aware perturbations ensuring high sensitivity to weight changes commonly introduced by attacks.

increases the likelihood of output differences across arbitrary downstream functions. This provides a task-agnostic mechanism for detecting tampering solely from black-box application responses.

To achieve this, we simulate tampering by perturbing the original model and optimize fingerprint twins to maximize the KL divergence between their similarity scores on clean and perturbed encoders—ensuring they remain highly consistent on the intact model yet strongly divergent after tampering. The added noise uses an *importance-aware perturbation* strategy that targets low-importance weights, estimated via layer-wise normalized gradients under a contrastive loss. This mirrors attacker behavior aimed at preserving task performance while modifying non-critical parameters, making even subtle unauthorized changes likely to produce detectable inconsistencies in application outputs. For verification, the two fingerprints in each twin are submitted to the application; if their output consistency deviates from the baseline expectation, tampering is flagged—without requiring ground-truth labels, internal access, or knowledge of the downstream pipeline.

Our main contributions are:

- *Black-box integrity verification for encoders:* We introduce a fingerprinting framework that verifies encoder integrity using only application outputs, without requiring internal access or knowledge of downstream pipelines.
- *Fingerprint twins:* We design paired fingerprints that stay highly consistent on the original encoder yet exhibit significant divergence after tampering, making output differences across arbitrary downstream tasks likely and detectable.
- *Importance-aware perturbation:* To emulate stealthy manipulations that preserve accuracy, we inject importance-weighted noise into encoder weights and optimize fingerprint twins to maximize KL divergence between clean and perturbed states, enhancing sensitivity to subtle changes.
- *Extensive evaluation:* Experiments across diverse datasets, encoder architectures, and tampering strategies demonstrate robust detection in strict black-box settings with negligible effect on standard functionality.

## 2 RELATED WORK

### 2.1 MODEL INTEGRITY VERIFICATION VIA FINGERPRINTING

Fingerprinting verifies model integrity without altering the original model by generating fingerprints highly sensitive to tampering and detecting changes through the model's responses (He et al., 2025). Existing techniques such as SSF (He et al., 2019), AID (Aramoon et al., 2021), PublicCheck (Wang

et al., 2022), MiSentry (He et al., 2024), IBSF (Bai et al., 2024), and SDBF (Bai et al., 2025b) focus on classifiers, while ESF (Bai et al., 2025a) targets LLM encoders. All these methods assume access to the model's predictions during verification. Applications embedding encoders—vision or language—do not expose such outputs. Integrity verification must therefore rely solely on final application outputs, while downstream pipelines are unknown when fingerprints are generated. These constraints render existing approaches inapplicable and motivate a task-agnostic fingerprinting framework that generalizes across encoder types.

## 2.2 ADVERSARIAL TAMPERING WITH PRE-TRAINED ENCODERS

Modern pre-trained encoders are typically obtained through self-supervised or contrastive learning on large-scale unlabeled data. While effective, these paradigms are highly vulnerable to adversarial manipulation.

**Poisoning and Backdoor Attacks.** Such training can be compromised even by a small proportion of malicious samples (Carlini & Terzis, 2021). PoisonedEncoder (Liu et al., 2022) and BadEncoder (Jia et al., 2022) embed backdoors during pre-training that persist across downstream tasks, while DRUPE (Tao et al., 2023) conceals poisoned data within the clean distribution to evade detection.

**Active Privacy Leakage Attacks.** Pre-trained models can also be manipulated to create severe privacy risks, such as amplifying membership inference or enabling fine-tuning data reconstruction—even under differential privacy (Wen et al., 2024; Feng & Tramèr, 2024).

Collectively, these threats underscore the need for robust mechanisms to ensure the integrity of pre-trained encoders before they are embedded into applications.

## 3 OUR METHOD: FINGERPRINT TWINS

The key idea of our method is to construct *fingerprint twins*—pairs of inputs that yield highly similar embeddings on an intact encoder but diverge significantly after any modification. This section first presents the threat model, then details the core designs and the algorithm for achieving this objective.

### 3.1 THREAT MODEL

We adopt a *white-box generation, black-box verification* paradigm, consistent with prior work (He et al., 2019). During fingerprint generation, full access to the encoder's parameters is assumed. Verification, however, occurs under a strict black-box setting: only application-level outputs—after unknown downstream processing—are observable, and encoder internals remain hidden. This aligns with the Encoder-as-a-Service (EaaS) deployment model, where encoders are embedded in proprietary pipelines, and verification must operate without task-specific assumptions.

Verification can be performed in private or public settings. In the public setting, fingerprints are distributed via a trusted third party for widespread integrity checks, at the risk of potential sample exposure.

### 3.2 DESIGN OBJECTIVES

We aim to develop a method that achieves the following three essential properties: **(i) Tampering-Agnostic:** Detects arbitrary encoder modifications without assuming specific attack types or objectives. **(ii) Downstream-Agnostic:** Requires no knowledge of downstream tasks, pipelines, or output semantics. **(iii) Stealthy:** Fingerprints resemble natural inputs, preventing detection by malicious providers attempting to evade integrity checks.

### 3.3 KEY COMPONENTS

#### 3.3.1 SIMULATING TAMPERING VIA IMPORTANCE-AWARE PERTURBATION

Let the encoder $f(\cdot; w)$ be parameterized by weights $w$. To emulate realistic manipulations such as backdoor injection or parameter fine-tuning, we introduce controlled perturbations that alter $w$ while

preserving primary functionality. The idea is to modify parameters with low functional importance, as these are the most likely targets for stealthy tampering. Importance is estimated by normalized and reversed layer-wise gradient magnitudes under a contrastive loss:

$$s_i = \frac{\max |\nabla w_i| - |\nabla w_i|}{\max |\nabla w_i| - \min |\nabla w_i|}, \tag{1}$$

where $\nabla w_i = \partial \mathcal{L}/\partial w_i$. Perturbations are then applied as:

$$\Delta w_i = \gamma \cdot s_i \cdot \mathcal{N}(0, \sigma^2), \tag{2}$$

where $\gamma$ controls perturbation strength. This strategy preserves task accuracy while altering internal representations in subtle yet detectable ways, effectively mimicking realistic attack goals—such as embedding a backdoor or applying model customization—without breaking the model's utility.

### 3.3.2 FINGERPRINT TWINS AND PAIRWISE DIVERGENCE OBJECTIVE

For a fingerprint twin $t = \{x_A, x_B\}$, define the similarity under encoder weights $w$ as:

$$S(t, w) = \text{sim}(f(x_A; w), f(x_B; w)), \tag{3}$$

where $\text{sim}(\cdot, \cdot)$ is cosine similarity. To ensure effectiveness across arbitrary downstream functions, we leverage the property that for any Lipschitz-continuous mapping, larger differences in encoder embeddings increase the likelihood of downstream output differences (Proposition 1). Based on this, we optimize twins to maximize:

$$\max_t \ S(t, w) \cdot \log\left(\frac{S(t, w)}{S(t, w + \Delta w)}\right), \tag{4}$$

which enforces high similarity on an intact model and a substantial relative drop after simulated tampering.

**Proposition 1** (Embedding vs. Output Divergence). *Let $f : \mathcal{X} \to \mathbb{R}^d$ be an encoder and $g : \mathbb{R}^d \to \mathcal{Y}$ an arbitrary downstream function. If $g$ is L-Lipschitz continuous with respect to a norm $\| \cdot \|$, then for any $\mathbf{x}_1, \mathbf{x}_2 \in \mathcal{X}$:*

$$\|g(f(\mathbf{x}_1)) - g(f(\mathbf{x}_2))\| \leq L \cdot \|f(\mathbf{x}_1) - f(\mathbf{x}_2)\|. \tag{5}$$

*Proof.* Since $g$ is $L$-Lipschitz continuous with respect to a norm $\| \cdot \|$, by definition of Lipschitz continuity, for any vectors $\mathbf{e}_1$ and $\mathbf{e}_2$, we have

$$\|g(\mathbf{e}_1) - g(\mathbf{e}_2)\| \ \leq \ L \|\mathbf{e}_1 - \mathbf{e}_2\| \quad \text{for all } \mathbf{e}_1, \mathbf{e}_2 \in \mathbb{R}^d.$$

Taking $\mathbf{e}_1 = f(\mathbf{x}_1)$ and $\mathbf{e}_2 = f(\mathbf{x}_2)$ for inputs $\mathbf{x}_1, \mathbf{x}_2 \in \mathcal{X}$, we obtain

$$\|g(f(\mathbf{x}_1)) - g(f(\mathbf{x}_2))\| \ \leq \ L \|f(\mathbf{x}_1) - f(\mathbf{x}_2)\|,$$

which immediately implies that larger separation in $\|f(\mathbf{x}_1) - f(\mathbf{x}_2)\|$ can cause a proportional increase in $\|g(f(\mathbf{x}_1)) - g(f(\mathbf{x}_2))\|$. This establishes the claim. $\qquad\square$

Proposition 1 implies that downstream output differences are bounded by encoder embedding differences. Increasing this bound—that is, enlarging the divergence between encoder embeddings—makes it more likely that downstream outputs will differ for any $L$-Lipschitz continuous downstream function. This insight motivates our design of fingerprint twins, which maintain high similarity on the intact encoder while inducing large divergence after perturbation.

### 3.4 FINGERPRINT GENERATION

Algorithm 1 summarizes the fingerprint-twin generation process. Starting from a clean sample (e.g., an image or audio clip), we initialize a perturbed pair within an $\epsilon$-bounded region and iteratively update it to maximize the KL divergence between their similarities on intact and tampered encoders:

$$\text{KL} = S(t, w) \log\left(\frac{S(t, w)}{S(t, w + \Delta w)}\right), \tag{6}$$

where $S(t, w)$ is the cosine similarity of twin $t = \{x_A, x_B\}$ under weights $w$. At each step, tampering is simulated via randomized importance-aware weight perturbations, and the twin is updated by gradient ascent and then projected back into the $\epsilon$-ball. This ensures the bound holds throughout optimization, keeping the fingerprints visually indistinguishable from the clean sample while being highly sensitive to weight changes.

---

**Algorithm 1** Fingerprint Twin Generation

---

**Input:** Encoder $f(\cdot; w)$, clean sample $x$, steps $N$, bound $\epsilon$, learning rate $\eta$
Compute layer-wise importance scores $s_i$
Initialize twin $t = \{x_A, x_B\}$ within $\epsilon$-ball of $x$
**for** $k = 1$ to $N$ **do**
    Sample $\gamma \in [\gamma_{\min}, \gamma_{\max}]$
    Perturb weights: $\Delta w_i = \gamma \cdot s_i \cdot \mathcal{N}(0, \sigma^2)$
    Compute KL: $S(t, w) \log \big(S(t, w) / S(t, w + \Delta w)\big)$
    Update $t \leftarrow t + \eta \nabla_t \mathrm{KL}$; project $t$ into $\epsilon$-ball
**end for**
**Return:** Optimized twin $t$

---

### 3.5 Hypothesis–Testing Based Tampering Detection

**Setup and notation.** After generating a collection of *fingerprint twin* probes (inputs designed so that an untampered model exhibits a *fingerprint twin* event), we test whether a downstream model has been tampered. For probe $i$, let $Y_i \in \{0, 1\}$ indicate a fingerprint twin (1) or not (0). Assuming i.i.d. probing under a fixed recipe, $Y_i \sim \mathrm{Bernoulli}(p)$, where $p$ is the *consistency rate*. Let $X = \sum_{i=1}^{n} Y_i$ be the total fingerprint twins observed in $n$ probes.

#### 3.5.1 Downstream–agnostic Calibration

From a calibration table built on reference models and multiple downstream datasets/attacks, we extract two operating points that yield *minimax* guarantees:

$$p_0^{\star} = \min_{\mathrm{rows}} \mathrm{Untampered}^{\uparrow}, \qquad p_1^{\star} = \max_{\mathrm{rows}} \max_{\mathrm{attacks}} \mathrm{Tampered}^{\downarrow}.$$

Here, $\mathrm{Untampered}^{\uparrow}$ is the observed fingerprint twin consistency for untampered models, and $\mathrm{Tampered}^{\downarrow}$ is the observed consistency under tampering. The pair $(p_0^{\star}, p_1^{\star})$ ensures error control across all downstream tasks considered, without assuming knowledge of the target dataset at test time.

#### 3.5.2 Minimax fixed–sample test

We perform an exact one–sided binomial test on the fingerprint twin consistency using the downstream–agnostic pair $(p_0^{\star}, p_1^{\star})$:

$$H_0 : p \geq p_0^{\star} \qquad \mathrm{vs} \qquad H_1 : p \leq p_1^{\star}, \quad \mathrm{with} \ p_1^{\star} < p_0^{\star}. \tag{7}$$

Given type–I and type–II error budgets $\alpha, \beta \in (0, 1)$ (e.g., $\alpha = \beta = 0.05$), we choose the smallest $n \in \mathbb{N}$ and an acceptance threshold $T \in \{0, 1, \ldots, n\}$ such that

$$\Pr_{X \sim \mathrm{Binom}(n, p_0^{\star})} \big[ X < T \big] \leq \alpha, \qquad \Pr_{X \sim \mathrm{Binom}(n, p_1^{\star})} \big[ X \geq T \big] \leq \beta. \tag{8}$$

**Decision rule.** Run $n$ fingerprint twin probes to obtain $X$. Declare *untampered* if $X \geq T$; otherwise, declare *tampered*. All probabilities in equation 8 are computed using exact binomial tails.

## 4 Experimental Evaluation

### 4.1 Experimental Setup

**Datasets.** We evaluate on four benchmarks spanning diverse downstream tasks and visual domains: CIFAR-10(Krizhevsky, 2009) and STL-10(Coates et al., 2011) for natural image classification, GT-SRB(Stallkamp et al., 2012) for traffic sign recognition, and SVHN(Netzer et al., 2011) for digit recognition from street-view imagery. This mix covers object categories at varying resolutions and scene structures—from general objects to structured symbols and digits—providing a broad test bed for transfer across heterogeneous tasks.

**Vision Encoders.** Unless stated otherwise, encoders are pre-trained with SimCLR (Chen et al., 2020) on either CIFAR-10 or STL-10 using a ResNet-18 backbone (He et al., 2016). We train for 1,000 epochs with Adam (initial learning rate $10^{-3}$).

**Downstream Classifiers.** Given a fixed pre-trained image encoder, we train a two-layer MLP (hidden sizes 512 and 256) on each downstream dataset's train split and report results on the test split. Optimization uses cross-entropy with Adam (50 epochs, initial learning rate $10^{-3}$).

**Tampering Attacks.** We consider three representative manipulations of encoders: **(i) BadEncoder** (Jia et al., 2022; Jia & collaborators, 2022), **(ii) DRUPE** (Tao et al., 2023; Tao & collaborators, 2023), and **(iii) INT8 quantization** (8-bit) using `Quanto` (Face, 2024).

**Fingerprint Twin Generation.** We generate 1,000 fingerprint-twin pairs per encoder using CIFAR-10 as the clean source. We set learning rate $\eta = 10^{-3}$ and perform 1,000 optimization steps. A binary search selects the weight-perturbation bound $\gamma$ in Eq. 2 to target an embedding-similarity band of 0.999–0.8 between untampered and perturbed encoders on clean images. The input-space budget is $\epsilon = 16/255$.

**Evaluation Metrics.** We report *clean accuracy* (standard test accuracy) and *consistency rate*, defined as the fraction of fingerprint-twin pairs whose predictions agree.

Table 1: Clean accuracy and consistency rate across different datasets and attacks.

| Pretraining | Downstream | Clean Accuracy | | | | Consistency Rate | | | |
| | | Untampered | Tampered | | | Untampered↑ | Tampered↓ | | |
| | | | BadEncoder | DRUPE | INT8-Quan | | BadEncoder | DRUPE | INT8-Quan |
|---|---|---|---|---|---|---|---|---|---|
| CIFAR-10 | CIFAR-10 | 91.03% | 90.88% | 90.72% | 89.48% | 98.00% | 25.00% | 24.10% | 22.00% |
| | STL-10 | 77.61% | 77.48% | 77.55% | 76.40% | 96.70% | 25.40% | 24.80% | 22.60% |
| | GTSRB | 79.84% | 80.49% | 79.79% | 72.55% | 84.10% | 18.70% | 14.90% | 14.70% |
| | SVHN | 62.02% | 70.20% | 71.59% | 58.40% | 83.80% | 24.80% | 27.90% | 21.00% |
| STL-10 | CIFAR-10 | 87.14% | 87.45% | 81.87% | 86.38% | 93.20% | 17.30% | 48.50% | 15.10% |
| | STL-10 | 81.03% | 80.53% | 74.70% | 79.89% | 94.10% | 16.20% | 52.90% | 14.00% |
| | GTSRB | 75.70% | 76.35% | 70.01% | 72.37% | 82.40% | 5.20% | 34.30% | 2.80% |
| | SVHN | 56.33% | 63.08% | 70.49% | 57.85% | 82.80% | 19.50% | 51.20% | 11.50% |

## 4.2 EVALUATION RESULTS

Table 1 reports the evaluation results across datasets and attacks. Several key conclusions emerge regarding the performance of tampered and untampered image encoders, particularly in terms of clean accuracy and consistency rate.

**(i) Clean Accuracy.** Untampered image encoders consistently achieve higher clean accuracy than their tampered counterparts across all datasets and attacks. For instance, with CIFAR-10 pretraining, untampered encoders reach 91.03% accuracy, while tampered versions (BadEncoder, DRUPE, and INT8-Quant) drop slightly to 89.48–90.88%. Similar trends hold for STL-10, GTSRB, and SVHN, indicating that tampering degrades encoder performance even without directly targeting the downstream classifier. However, the reduction is typically less than 2%, making detection by clean accuracy alone difficult.

**(ii) Consistency Rate.** Consistency rates, measured using fingerprint twins, reveal a sharp contrast between untampered and tampered encoders. Untampered encoders maintain consistently high rates, while tampered encoders show substantial declines. For example, on CIFAR-10, the untampered encoder achieves 98.00%, compared to 25.00%, 24.10%, and 22.00% for BadEncoder, DRUPE, and INT8-Quant, respectively. The same pattern is observed on STL-10, where the untampered encoder scores 96.70% versus 22.60–25.40% for tampered models. These results demonstrate the effectiveness of fingerprint twins in detecting tampering, as compromised encoders fail to sustain high consistency.

**(iii) Downstream-agnostic integrity decisions (minimax fixed–$n$).** Aggregating over Table 1, the downstream-agnostic calibration yields $p_0^\star = \min_{\text{rows}} \text{Untampered}^\uparrow = 0.824$ and $p_1^\star = \max_{\text{rows,attacks}} \text{Tampered}^\downarrow = 0.529$. Using the one-sided binomial design in §3.5, we obtain a unified decision policy that simultaneously controls type I and type II errors across all datasets and attacks. For $\alpha = \beta = 0.05$, the policy requires only $n = 28$ fingerprint twin probes with threshold $T = 20$ (accept untampered if $X \geq 20$); for $\alpha = \beta = 0.01$, $n = 53$ with $T = 37$. As summarized in Table 2, this approach cleanly separates untampered ($\geq p_0^\star$) from tampered ($\leq p_1^\star$) encoders, explaining the small probe budgets.

**(iv) Clean-accuracy tests are sample-inefficient; the minimax fixed–$n$ policy is superior.** Table 3 shows that two-sided accuracy-difference tests with $\alpha = \beta = 0.05$ demand prohibitively large sample sizes due to minimal per-row differences (e.g., $|\Delta| \leq 0.65$ percentage points in six of eight rows). Required probe counts reach $n \approx 4.75 \times 10^5$–$8.37 \times 10^6$ for CIFAR→{CIFAR, STL, GTSRB}, and still $n \approx 5.6 \times 10^4$–$1.5 \times 10^5$ for STL rows; only the larger SVHN drop ($-3.62$ points) reduces the need to $n \approx 2.37 \times 10^3$. By contrast, the downstream-agnostic minimax fixed–$n$ test based on fingerprint consistency achieves the same error guarantees with only $n = 28$ probes at $\alpha = \beta = 0.05$ (or $n = 53$ at $\alpha = \beta = 0.01$)—a reduction of $1.7 \times 10^4$–$3.0 \times 10^5$ probes for the hardest rows and $85\times$ fewer even for SVHN. Moreover, the single $(n, T)$ policy in Table 2 applies uniformly across datasets and attacks without downstream labels, underscoring the practical advantage of the minimax fixed–$n$ method for integrity auditing.

Table 2: Downstream–agnostic (minimax) fixed–$n$ hypothesis–testing results derived from Table 1. For consistency we use the fingerprint twin consistency rate. Minimax parameters are $p_0^\star = \min_{\text{rows}} \text{Untampered}^\uparrow$, $p_1^\star = \max_{\text{rows, attacks}} \text{Tampered}^\downarrow$. The decision rule is: accept *untampered* if $X \geq T$ after $n$ probes.

| Metric | $p_0^\star$ | $p_1^\star$ | $\alpha$ | $\beta$ | $n$ (min) | $T$ | Decision rule |
|---|---|---|---|---|---|---|---|
| Consistency (fingerprint twin) | $p_0^\star = 0.824$ | $p_1^\star = 0.529$ | 0.05 | 0.05 | **28** | **20** | accept if $X \geq 20$ |
|  | $p_0^\star = 0.824$ | $p_1^\star = 0.529$ | 0.01 | 0.01 | **53** | **37** | accept if $X \geq 37$ |

**Notes.** Values computed from Table 1: For consitency rate,

$p_0^\star = \min\{0.980, 0.967, 0.841, 0.838, 0.932, 0.941, 0.824, 0.828\} = 0.824$;

$p_1^\star = \max\{0.25, 0.254, 0.187, 0.248, 0.173, 0.162, 0.052, 0.195, 0.241, 0.248, 0.149, 0.279, 0.485, 0.529,$
$0.343, 0.512, 0.22, 0.226, 0.147, 0.21, 0.151, 0.14, 0.028, 0.115\} = 0.529$.

Table 3: Per–row fixed–$n$ hypothesis–testing results using **clean accuracy** with a *two–sided accuracy–change* test ($H_0 : A = A_0$ vs $H_1 : A \neq A_0$). For each row, $A_0$ is the untampered accuracy and $A_1^\star$ is the *closest* tampered accuracy (across BadEncoder/DRUPE/INT8-Quan) to yield the hardest detectable change. Sample sizes $n$ are planning–level normal–approximation results for $\alpha = \beta = 0.05$ (two–sided $\alpha$).

| Pretraining | Downstream | $A_0$ (Untampered) | $A_1^\star$ (Closest Tampered) | $\Delta = A_1^\star - A_0$ | $\alpha$ | $\beta$ | $n$ (approx) |
|---|---|---|---|---|---|---|---|
| CIFAR–10 | CIFAR–10 | 91.03% | 90.88% (BadEncoder) | $-0.15\%$ | 0.05 | 0.05 | **474,817** |
|  | STL–10 | 77.61% | 77.55% (DRUPE) | $-0.06\%$ | 0.05 | 0.05 | **6,277,880** |
|  | GTSRB | 79.84% | 79.79% (DRUPE) | $-0.05\%$ | 0.05 | 0.05 | **8,373,451** |
|  | SVHN | 62.02% | 58.40% (INT8–Quan) | $-3.62\%$ | 0.05 | 0.05 | **2,369** |
| STL–10 | CIFAR–10 | 87.14% | 87.45% (BadEncoder) | $+0.31\%$ | 0.05 | 0.05 | **150,100** |
|  | STL–10 | 81.03% | 80.53% (BadEncoder) | $-0.50\%$ | 0.05 | 0.05 | **80,627** |
|  | GTSRB | 75.70% | 76.35% (BadEncoder) | $+0.65\%$ | 0.05 | 0.05 | **56,101** |
|  | SVHN | 56.33% | 57.85% (INT8–Quan) | $+1.52\%$ | 0.05 | 0.05 | **13,780** |

**Notes.** The two–sided planning formula (one–sample proportion) is

$n \approx \left(z_{1-\alpha/2}\sqrt{A_0(1 - A_0)} + z_{1-\beta}\sqrt{A_1^\star(1 - A_1^\star)}\right)^2 / (A_0 - A_1^\star)^2$, with $z_{1-\alpha/2} = 1.96$ and $z_{1-\beta} = 1.645$ for $\alpha = \beta = 0.05$. Values are rounded; exact binomial designs will differ slightly. Because several tampered accuracies are extremely close to $A_0$, two–sided change detection can require very large $n$, underscoring why consistency (Table 2) is the preferred downstream–agnostic signal.

## 4.3 ABLATION STUDY

**Importance-aware perturbation.** Table 4a reports the change ($\Delta$) in consistency after applying importance-aware perturbations to fingerprint twins. Untampered models improve by $2.0\%$–$6.0\%$ across tasks, while tampered models decline (up to $11.4\%$), widening the decision margin.

**Varying input budget $\epsilon$.** Increasing $\epsilon$ from $4/255$ to $16/255$ raises untampered consistency ($96.8\% \rightarrow 98.0\%$) while suppressing tampered consistency (e.g., BadEncoder $47.6\% \rightarrow 25.0\%$), yielding large, monotonic margins (Table 4b).

**Varying weight-perturbation bound $\gamma$.** Lowering $\gamma$ (more aggressive perturbations on low-importance weights) increases untampered consistency and further depresses tampered consistency (Table 4c).

Table 4: Ablations on importance-aware perturbation, input-perturbation magnitude $\epsilon$, weight-perturbation bound $\gamma$, and tampering magnitude (CIFAR-10).

(a) Importance-aware perturbation.

| Dataset | Unt.↑ | BadEnc.↓ | DRUPE↓ | INT8↓ |
|---------|-------|----------|--------|-------|
| CIFAR-10 | +5.10 | -2.80 | -4.00 | -9.80 |
| STL-10 | +6.00 | -4.90 | -4.60 | -8.50 |
| GTSRB | +2.00 | -3.40 | -10.30 | -8.20 |
| SVHN | +2.30 | -6.50 | -11.40 | -10.10 |

(b) Effect of $\epsilon$.

| $\epsilon$ | Unt.↑ | BadEnc.↓ | DRUPE↓ | INT8↓ |
|------------|-------|----------|--------|-------|
| 4/255 | 96.8 | 47.6 | 46.4 | 27.4 |
| 8/255 | 97.6 | 39.6 | 39.3 | 25.6 |
| 16/255 | 98.0 | 25.0 | 24.1 | 22.0 |

(c) Effect of $\gamma$.

| $\gamma$ | Unt.↑ | BadEnc.↓ | DRUPE↓ | INT8↓ |
|----------|-------|----------|--------|-------|
| 0.9 | 91.6 | 30.7 | 28.9 | 15.2 |
| 0.8 | 98.0 | 25.0 | 24.1 | 22.0 |

(d) Subtler tampering (1 epoch SimCLR).

| Downstream | CIFAR | STL10 | GTSRB | SVHN |
|------------|-------|-------|-------|------|
| $10^{-3} \times 1$ | 44.0 | 39.6 | 22.9 | 30.4 |
| $10^{-4} \times 1$ | 75.9 | 69.4 | 64.0 | 67.4 |

**Subtle Tampering.** We also simulate mild post-deployment drift by fine-tuning the CIFAR-10–pretrained encoder for one epoch using SimCLR with learning rates $10^{-3}$ and $10^{-4}$. Consistency rises relative to stronger attacks but remains well below untampered levels (Table 4d), preserving a significant margin across tasks.

## 4.4 FURTHER ANALYSIS

**Model susceptibility to baseline noisy samples.**

To isolate the impact of perturbation *magnitude* from that of our structure-aware fingerprint twins, we conduct a control experiment in which the twins are replaced by i.i.d. Gaussian noise whose $\ell_\infty$ norm matches the budget $\epsilon$. As summarised in Table 5, random noise fails to create any meaningful separation between untampered and tampered encoders. On CIFAR-10 and STL-10, every encoder—whether pristine or tampered by BadEncoder, DRUPE, or INT8-Quantization—retains at least 94% consistency. Even on the more challenging GTSRB and SVHN datasets, consistency never drops below 75%, and the largest observed gap (INT8-Quantization on SVHN) is a negligible 3.2%.

These results demonstrate that merely injecting noise of the correct scale is insufficient for integrity verification; it neither perturbs salient features nor exposes tampering-induced representation drift. In contrast, our structure-aware fingerprint twins are explicitly aligned with feature importance, yielding substantial divergence in tampered models while remaining stable in untampered ones. Hence, *structure-aware twin construction—rather than perturbation magnitude alone—is essential for reliable detection*.

Table 5: Random-noise baseline: consistency (%).

| Downstream | Untampered ↑ | BadEncoder ↓ | DRUPE ↓ | INT8-Quantization ↓ |
|------------|--------------|--------------|---------|---------------------|
| CIFAR10 | 94.5 | 95.5 | 95.5 | 94.7 |
| STL10 | 94.9 | 95.3 | 93.8 | 91.4 |
| GTSRB | 77.7 | 81.4 | 82.0 | 76.6 |
| SVHN | 80.5 | 84.2 | 84.5 | 77.3 |

**Downstream complexity and real-world use cases.**

Table 6 probes how the size of the downstream label space influences our integrity signal. Moving from CIFAR-10 (10 classes) to the considerably more fine-grained CIFAR-100 (100 classes) naturally *lowers* the consistency of untampered encoders from 98.0% to 82.2% (a 15.8-pp drop), owing to harder decision boundaries and noisier gradient updates. Crucially, all three tampering mechanisms suffer a far steeper collapse: BadEncoder falls from 25.0% to 13.0%, DRUPE from 24.1% to 12.1%, and INT8-Quantisation from 22.0% to 6.9%. Hence, even with harder tasks and noisier gradients, the twin fingerprints remain discriminative.

Table 6: Impact of downstream complexity.

| Downstream | Untampered ↑ | BadEncoder ↓ | DRUPE ↓ | INT8-Quantization ↓ |
|---|---|---|---|---|
| CIFAR100 | 82.2 | 13.0 | 12.1 | 6.9 |
| CIFAR10 | 98.0 | 25.0 | 24.1 | 22.0 |

**Evaluation on Larger Dataset Pretrained Models.** To verify that our findings are not an artefact of any specific backbone, we replicate the experiment on an stronger *ImageNet–pre-trained ResNet-18*. The results, summarised in Table 7, are fully consistent with earlier observations. Across three downstream datasets, the untampered encoder remains highly consistent—87.50% on CIFAR-10, 89.10% on STL-10, and 77.20% on GTSRB. In stark contrast, BadEncoder tampering drives the metric down to single-digit (3.75%) consistency on CIFAR-10 and GTSRB (9.00%), with only a modest 18.90% on STL-10.

Table 7: Fingerprint detection on ImageNet-pretrained ResNet-18.

| Downstream | Untampered ↑ | BadEncoder ↓ |
|---|---|---|
| CIFAR10 | 87.50 | 3.75 |
| STL10 | 89.10 | 18.90 |
| GTSRB | 77.20 | 9.00 |

**Evaluation on Large-scale or Higher-resolution Encoders.** To highlight the effectiveness of our method on large-scale or higher-resolution encoders, we evaluated our method on a cutting-edge unlabeled large-scale images pretrained DINOv2( Oquab et al. (2023)) Vision Transformer model (ViT-Base) for the downstream ImageNet classification task. As shown in Table 8, the consistency rate reached 97.9% for the untampered encoder and dropped significantly to 71.3% after compression (via INT8 quantization), while downstream accuracy remained almost unchanged (from 83.414% to 83.294%). This demonstrates that even subtle parameter-level tampering, such as compression, on large-scale encoders can be reliably detected. Although downstream performance degrades minimally, the fingerprint twins amplify the change in consistency rate (around 27% drop), enabling detection with only a small number of queries. Specifically, just 15 fingerprint twins were sufficient to detect tampering, with both Type I and Type II error rates remaining below 5%.

Table 8: Fingerprint detection on ViT base encoder.

| Tampering | Clean Accuracy | Consistency Rate |
|---|---|---|
| Untampered | 83.414% | 97.9% |
| INT8-Quan | 83.294% | 71.3% |

**Evaluation on Speech Encoder.** To demonstrate effectiveness on non-vision modalities, we evaluated the Wav2Vec 2.0 Base(Baevski et al. (2020)) encoder, which is widely used as a general-purpose speech encoder, on the downstream Automatic Speech Recognition (ASR) task. We generated 200 fingerprint twin pairs, defining consistency as having a Word Error Rate (WER) between twin outputs below a predefined threshold of 0.20, which was determined to keep low false-alarm while maintain high detection rate. As shown in Table 9, comparing the original encoder with an INT8-quantized version reveals that while standard ASR performance remains nearly unchanged, the fingerprint consistency rate drops sharply from 77.5% to 39.5% and the mean WER between twin outputs more than doubles from 12.50% to 30.46%. This significant divergence in downstream token sequences confirms that parameter tampering causes detectable output variations, proving that our method effectively generalizes to speech encoders.

**Computational Overhead of White-box Fingerprint Generation.** To demonstrate the low overhead of our method in white-box generation, we measured the average time cost of generating a single fingerprint twin on a single NVIDIA A100 GPU across two representative settings. For a small ResNet-18 model, generating one twin pair on CIFAR10 takes only 0.41 seconds, while for a large ViT-based encoder, generating a twin on ImageNet requires approximately 40 seconds. Since fingerprint generation is a one-time, offline process performed by the model owner before deployment, this overhead remains minimal, even for foundation-model-scale encoders.

**Evaluation on Sensitivity to Encoder Fine-Tuning.** To demonstrate sensitivity to subtle encoder fine-tuning changes, we generated fingerprint twins on converged baseline checkpoint trained on an encoder(MoCov3+ResNet-18) on CIFAR-10. These twins were then evaluated on a further fine-tuned encoder, where the encoder performance remained largely unchanged, with clean accuracy fluctuating within 1%. As shown in Table 10, across multiple downstream tasks (CIFAR-10,

Table 9: Detection on speech encoder.

| Metric | Untampered | Tampered |
|--------|------------|----------|
| Twin-pair consistency rate | 77.5% | 39.5% |
| mean WER - clean samples | 3.95% | 4.32% |
| mean WER - FP twins | 12.50% | 30.46% |

Table 10: Effect of Fine-Tuning Encoder.

| Downstream | Untampered ↑ | Finetuned ↓ |
|------------|--------------|-------------|
| CIFAR-10 | 95.4 | 45.0 |
| STL-10 | 95.7 | 55.4 |
| SVHN | 92.4 | 61.9 |
| GTSRB | 89.1 | 46.1 |

STL-10, SVHN, GTSRB), we observed the fingerprint twins' consistency rate sharply dropped from 89.1%–95.7%(untampered) to 45%–61.9%(fine-tuned). This indicates that fingerprint twins strongly amplify subtle weight-level modifications and can reliably detect fine-tuning. Despite negligible change in pretrained encoder utility, such a large drop in consistency demonstrates that fingerprint twins are capable of amplifying subtle weight-level pretrained encoder modifications into large and detectable discrepancies. In other words, even minor fine-tuning or small parameter perturbations that would be invisible to users relying solely on task performance are reliably exposed by our method. Feature-space transformations, including Channel-wise scaling, doesn't impact model weights directly, and is out of scope of our discussed threat model.

**Adaptive Attacker.** We consider an adaptive variant of BadEncoder where the attacker adds an extra adversarial loss term encouraging the tampered encoder to maintain small embedding deviations for selected samples while still introducing the backdoor. Under the encoder with adversarial loss scenario, we evaluate fingerprint-twin consistency rate across multiple downstream tasks. As shown in Table 11, even though the attacker explicitly optimizes to keep embeddings close, we still observe marked deviations in consistency compared to the untampered baseline. Hypothesis testing on these consistency rates continues to reliably flag tampering. Our fingerprint twins remain effective even under an adaptive attacker aware of the detection mechanism, further supporting the robustness of the proposed approach.

Table 11: Impact of Adaptive Attacker.

| Downstream | Untampered | BadEncoder Consistency Rate | Adaptive Consistency Rate |
|------------|------------|----------------------------|---------------------------|
| CIFAR-10 | 98.00% | 25.00% | **31.10%** |
| STL-10 | 96.70% | 25.40% | **32.30%** |
| GTSRB | 84.10% | 18.70% | **25.60%** |
| SVHN | 83.80% | 24.80% | **32.10%** |

## 5 CONCLUSION

We presented a fingerprinting framework for black-box integrity verification of pre-trained encoders embedded within unknown downstream pipelines. Our approach is grounded in a theoretical insight (Proposition 1): under mild assumptions, increasing encoder embedding divergence enlarges the bound on downstream output differences for two inputs, thereby making the application's responses to a fingerprint pair more likely to diverge under tampering. Building on this principle, we introduce *fingerprint twins*, designed to remain highly similar on an intact encoder yet diverge under tampering, enabling detection based solely on application outputs. We simulate realistic manipulations using *importance-aware perturbations* and optimize twins to maximize KL divergence between clean and perturbed states while constraining perturbations within an $\epsilon$-ball to preserve input naturalness. This combination provides task-agnostic, unobtrusive integrity verification without requiring internal access or ground-truth labels. Extensive experiments confirm robustness across architectures, datasets, and tampering strategies with minimal impact on system functionality, offering a practical path toward securing shared AI infrastructures under strict black-box constraints.

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

## LLM USAGE STATEMENT

We used a large language model (LLM) strictly for editorial assistance. The LLM was employed to correct grammar and spelling, improve clarity and concision at the sentence/paragraph level, standardize terminology and style, and suggest alternative phrasings. It was **not** used to generate research ideas, design experiments, analyze data, produce code, create figures, draft technical content (e.g., methods, results), or write literature reviews. All model outputs were reviewed and edited by the authors, who take full responsibility for the final text and for verifying all claims and citations. No confidential or unpublished data were provided to the LLM beyond text written by the authors.

# A APPENDIX

## A.1 PERCEPTUAL QUALITY

Figure 2 shows representative twins on CIFAR-10. Despite structure-aware perturbations, the twins are visually indistinguishable from clean images (global color, contours, and textures are preserved), indicating that our fingerprints do not compromise perceptual quality.

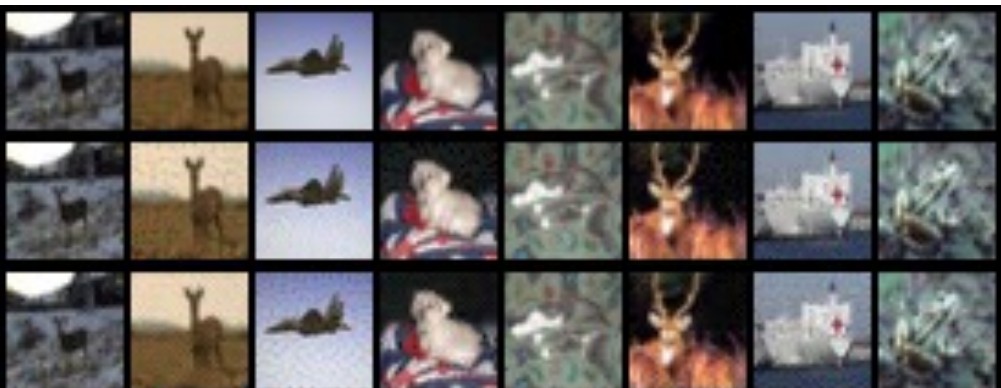

Figure 2: Perceptual quality of fingerprint twins generated on CIFAR-10.

