# OpenReview forum: "Tampering Detection for Pre-trained Encoders Using Fingerprintwins"
_ICLR.cc/2026/Conference — Submitted to ICLR 2026_

### Official Review · Reviewer_qS2m · 2025-10-29

**Soundness:** 1
**Presentation:** 3
**Contribution:** 2
**Rating:** 2
**Confidence:** 4

**Summary:**

This work presents a novel method of discovering if pre-trained general encoder weights were changed after the pre-training. It is a hard problem, since during inference the encoder is hidden and an unknown downstream pipeline follows the encoding. The authors propose an fingerprinting scheme based on twined inputs that helps to discover such changes (adversarial or benign). The method is tested with importance-aware perturbations used to mimic adversarial weights tampering.

**Strengths:**

In general I have a very positive view on this work:
- it is well-organized and the main idea is clearly presented.
- the work shows interesting insights in small scale like ablation study on subtle tampering or additional control experiment with noisy samples.
- the problem of tampering detection seems interesting.

**Weaknesses:**

Unfortunately, I have big concerns connected to the experimental setup (W1/W4) and questions about the usefulness of the threat model proposed (W2) and how the tampering mimicking generalize (W3). Those are followed by other issues (W5).

W1. Resnets pretrained on CIFAR-10 are not up to the task of being a general purpose pre-trained encoders even for small tasks used in this work.  Evaluation should use bigger encoders pre-trained using at least Imagenet to be used as a general purpose for given downstream tasks. (only partial Tab7 results). Most of the experiments cannot be considered real-world use cases (line 445).


W2. I wonder what are the possible use cases behind proposed/ used threat model. It seems odd to me that non-adversarial party has white access to the model at first - fingerprinting generation, and then only black box during verification. This is an important point as it could change the applicability of the method.

W3. Applicability of the proposed method and how it generalizes. Is mimicking via I-A perturbation enough to support your claims? E.g. could you explain if small fine-tuning or adding an adapter layer of the encoder would be considered tampering? (Explain in detail experiments like tab 4d.) What is allowed in the downstream pipeline and what is not?

W4. Tab 1 results: some tampered encoders have better performance than untampered ones, some have much worse - to my understanding it shows that the evaluation setup is lacking (see W1). Could you comment on that?

W5. Others:
- Could you write more about limitations of the proposed method?
- Tested tampering attacks are not described or motivated.

**Questions:**

In general: weaknesses, especially W1.

Ad W2.
- Could you elaborate on the use cases?
- How to understand the assumption of trusted third party widespread integrity checks unknowingly to adversaries?
- Could you elaborate on how general is the proposed method? In particular assumptions (lines 162-163) using importance-aware perturbation and the motivation for this in previous works?
- The models utility for specific downstream task could be improved by fine-tuning, would it be detected as tampering (which would mean that your method hinder the utility)?

Ad W2/W4
- What would happen if the attacker would be aware of the fingerprinting used?

---

> ### Author Response · Authors · 2025-12-03
>
> We thank the reviewer for the thoughtful and detailed comments. Below we address each main concern in turn and clarify how our method:
> 1. operates on realistic, general-purpose encoders,
> 2. is grounded in a standard and practical threat model,
> 3. correctly flags fine-tuning and other parameter-level changes as tampering,
> 4. uses a general tampering-mimicking mechanism, and
> 5. evaluates representative tampering attacks, including an adaptive attacker.
>
> ---
>
> ### W1 – Are we using “realistic, general-purpose” encoders?
>
> To emphasize that our method targets realistic, general-purpose encoders, we evaluate on **widely used, large-scale pretrained models in both vision and audio**:
> - Vision domain.
> We use DINOv2 ViT-B/14, a large-scale pretrained vision transformer that is widely regarded as a strong, general-purpose visual feature extractor. It is commonly used far beyond CIFAR-10, including various downstream classification including ImageNet, detection, and retrieval tasks.
> - Audio domain.
> We use Wav2Vec 2.0, a self-supervised speech encoder pretrained on large unlabeled speech corpora (LibriSpeech LS-960 and LibriVox LV-60k). Wav2Vec 2.0 is a standard general-purpose speech encoder in the community.
>
> **In both settings, our fingerprint-twins method reliably detects subtle parameter-level tampering that leaves downstream accuracy almost unchanged, yet induces large drops in consistency rate.** Experiments are described below.
>
> **1. Vision: Large-Scale Encoder (DINOv2-ViT-B/14, ImageNet)**
>    - Encoder: pretrained DINOv2-ViT-B/14 vision encoder.
>    - Fingerprint generation: 1000 image twin pairs generated on the clean encoder.
>    - Downstream task: ImageNet classification.
>    As shown in the following results, although INT8 compression reduces downstream accuracy by only 0.1%, the consistency rate drops by ~27% (97.9% → 71.3%). This demonstrates that fingerprint twins are highly sensitive to subtle parameter-level tampering, even on large-scale, general-purpose encoders.
>
> | Model / Metric           |  Clean Accuracy                      | Consistency Rate |
> | --------------------------- | ------------------------------------ | ---------------- |
> | Untampered model   | 83.4%                              | 97.9%            |
> | Tampered model (INT8 quantization) | 83.3%                              | 71.3%            |
>
> **2. Audio: Speech Encoder (Wav2Vec-Base–LibriSpeech)**
>   - Encoder: pretrained Wav2Vec-Base speech encoder.
>   - Fingerprint generation: 200 audio twin pairs on the clean encoder.
>   - ASR downstream: linear projection + CTC decoding to produce transcripts.
>   - Consistency criterion: word-error rate (WER) between twin outputs < 0.20.
>   As shown in the following results, the ASR performance on clean inputs is essentially unchanged by tampering (mean WER 3.95% vs. 4.32%). In contrast, twin-pair consistency drops sharply from 77.5% to 39.5%, and the mean WER between twin outputs more than doubles (12.50% → 30.46%).
>
> | Condition                           | Untampered    | Tampered           |
> | ----------------------------------- | ------------- | ------------------ |
> | Twin‑pair consistency rate          | 77.5%     | 39.5%         |
> | mean WER - clean samples            | 3.95%         | 4.32%              |
> | mean WER - FP twins                 | 12.50%        | 30.46%             |
>
>
> These experiments show that our method applies naturally to general-purpose encoders in both vision and speech, and that the core mechanism, embedding instability translating into large divergences in downstream outputs, holds across modalities under realistic encoder-as-a-service scenarios.

---

> ### Author Response · Authors · 2025-12-03
>
> ### W2 & Ad W2(the 1st and 2nd question) – Threat model and use-case realism
>
> Our setting is **closely aligned with existing standard DNN tampering-detection literature [c–h]** and reflects a natural cloud encoder-as-a-service (EaaS) deployment with three roles:
>
> 1. Model owner / vendor.
> Holds the pretrained encoder and has white-box access before deployment. The owner uses this access once to generate fingerprint twins on the clean encoder.
> 2. Trusted third party (TTP).
> Securely stores/distributes fingerprint twins and orchestrates periodic integrity checks[n]. As in prior work, we explicitly allow that the cloud service provider may be untrusted or malicious, e.g., performing backdoor injection or compression to reduce cost. These are precisely parameter-level tampering actions that break the integrity of the originally certified encoder.
> 3. End-users.
> Interact with the system only via black-box API calls to the deployed downstream tasks (classification, ASR, etc.).
>
> We follow the standard white-box generation, black-box verification paradigm:
> - Fingerprint generation (white box).
> The model owner uses full access to the clean encoder and normal data to synthesize fingerprint twins. This is done once before release and defines the integrity baseline.
> - Tampering detection (black box).
> After deployment, only black-box access to the downstream task is assumed. Verifiers send queries and observe outputs (e.g., hard labels, probabilities, or transcripts), but cannot inspect or download model weights, which matches realistic EaaS constraints.
>
> This setting is not ad-hoc. It is consistent with and explicitly motivated by prior work on strict black-box integrity verification [c–h]. Our method is specifically designed to operate under these realistic deployment constraints.
>
> ---
>
> ### W3 & Ad W2(the 4th question) - Will encoder and downstream fine-tuning be flagged as tampering?
>
> As stated clearly in Sec. 3.1 (Threat Model), our goal is to verify the integrity of the encoder itself, independent of the downstream task. By generating fingerprint twins on the encoder, our method ensures consistency across any unknown downstream task. For an untampered encoder, the fingerprint twins remain consistent across different downstream tasks, while for a tampered encoder, they exhibit noticeable inconsistencies.
>
> Any modification to the encoder, including fine-tuning, is particularly challenging because we cannot foresee the specifics of the downstream task. Therefore, our method verifies the encoder’s integrity by checking whether the fingerprint twins remain consistent, regardless of the downstream task. This approach does not depend on any particular downstream task but ensures the encoder’s integrity in the face of unknown tasks.
>
> Regarding the downstream pipeline, our method allows modifications to the downstream layers (e.g., adding adapter layers, task-specific networks, etc.), as long as the core encoder remains unchanged. From the standpoint of integrity:
> > **Any modification of encoder weights**, including fine-tuning, constitutes tampering, even if it improves some downstream performance.
>
> This distinction is critical, as our method is designed to detect changes to the encoder itself, not the downstream task components.
>
> To ensure rigor, we conducted encoder fine-tuning experiments. We trained an encoder from scratch using MoCov3 with a ResNet-18 backbone on CIFAR-10, selecting the converged checkpoint as the untampered encoder. Fingerprint twins were then generated from this checkpoint. We further fine-tuned the converged encoder and evaluated both the untampered and fine-tuned encoders for accuracy and fingerprint-twin consistency across multiple downstream datasets. The consistency rate results are:
>
> |   Dataset     | Untampered |  Fine-tuned  |
> | :-----------: | :--------: | :---:        |
> |   CIFAR‑10    |    95.4    | 45.0         |
> |   STL‑10      |    95.7    | 55.4         |
> |   SVHN        |    92.4    | 61.9         |
> |   GTSRB       |    89.1    | 46.1         |
>
> Across multiple downstream tasks (CIFAR‑10, STL‑10, SVHN, GTSRB), the encoder’s utility remains essentially unchanged (downstream accuracy varies by less than 1%, and even raises a litlle bit), yet the consistency rate of fingerprint twins drops sharply—from approximately 89.1–95.7% in the untampered setting to 45.0–61.9% after fine-tuning. This provides clear evidence that fingerprint twins amplify subtle weight-level modifications into large, easily detectable discrepancies.
>
> As described in Sec. 3.1, our threat model allows arbitrary downstream pipelines (e.g., linear heads, task-specific networks, non-neural post-processing), provided they are built on top of a fixed, certified encoder. Changing only the downstream head is allowed, while changing the shared encoder weights is tampering. This is exactly the notion of integrity that the reviewer is probing, and our method is designed to detect such encoder-level changes.

---

> ### Author Response · Authors · 2025-12-03
>
> ### Ad W2(the 3rd question) – Is importance-aware perturbation a general tampering-mimicking method?
> In Sec. 3.3.1, we introduce importance-aware perturbation as a generic way to simulate tampering during fingerprint generation. This choice is motivated by how realistic attackers operate:
> - In realistic tampering (backdoor injection, compression, etc.), attackers usually **aim to preserve downstream utility**, so they cannot arbitrarily corrupt all weights.
> - To keep performance high, they will **avoid significantly perturbing high-importance weights**, which strongly affect the encoder’s behavior, and instead target lower-importance weights that can be modified with less impact on utility.
>
> Our strategy, perturbing low-importance parameters more, is therefore aligned with realistic tampering motivations: **maintain utility by keeping high-importance parameters stable, while still introducing parameter-level changes via low-importance ones.**
>
> Empirically, across multiple tampering mechanisms (BadEncoder, DRUPE, quantization), modalities (vision and audio), and in the ablation study on importance-aware perturbation (Sec. 4.3), we observe that importance-aware perturbations consistently yield fingerprint twins that robustly distinguish between tampered and untampered encoders. This indicates that our perturbation scheme is not tailored to a narrow artificial scenario, but rather **captures a general and plausible model-tampering pattern**.
>
> ---
>
> ### W4 – Some tampered encoders show better clean accuracy than clean ones — is the evaluation invalid?
>
> This phenomenon is in fact **well-known and documented in the backdoor literature**. In particular, the original BadEncoder paper [i] explicitly reports that backdoored encoders can exhibit equal or even higher clean accuracy than their clean counterparts on certain downstream tasks, while still enabling backdoor behavior on triggered inputs. This aligns with the attacker’s goal: maintain or even slightly improve clean accuracy to remain stealthy while inserting malicious behavior.
>
> Our experiments faithfully reproduce this behavior **using the official implementation and default hyperparameters** [j]. Therefore, the observed improvement in clean accuracy for some tampered encoders is not an artifact of our evaluation, but rather a known characteristic of realistic backdoor attacks on pretrained encoders. Our results **being consistent with the original BadEncoder paper** strengthens, rather than weakens, the validity of our experimental setup.
>
> ---
>
> ### W5 – Insufficient description / motivation of tampering attacks
>
> Due to space constraints, the main paper briefly references the tampering attacks in Sec. 4.1. Here we clarify that our evaluation is based on three representative, well-motivated tampering mechanisms:
>
> 1. BadEncoder [i].
> BadEncoder introduces a backdoor into pretrained encoders so that any downstream classifier built on top can be manipulated on trigger-containing inputs while preserving clean accuracy. We use the official implementation and default hyperparameters [j].
>
> 2. DRUPE [k].
> DRUPE enhances backdoor attacks by embedding poisoned samples within the in-distribution data, making them harder to detect via traditional OOD-based defenses. We again use the official implementation with default hyperparameters [l].
>
> 3. Integer quantization (compression) [m].
> INT8 quantization is a widely used practical technique to reduce model size and inference cost by mapping floating-point weights/activations to lower-precision integers. It is routinely deployed in large-scale systems and can significantly alter encoder parameters while preserving accuracy. We use the popular PyTorch-based backend optimum-quanto [m].
>
> All three attacks share a unifying characteristic: they modify encoder parameters while carefully preserving utility on clean data, either to remain stealthy (backdoor attacks) or to improve cost-efficiency (compression). Together, they span a **broad and realistic spectrum** of parameter-level tampering scenarios in real-world encoder-as-a-service deployments.

---

> ### Author Response · Authors · 2025-12-03
>
> ### Ad W2/W4 – Adaptive attacker
> We consider an adaptive variant of BadEncoder in which the attacker introduces an additional adversarial loss term explicitly encouraging the tampered encoder to keep embeddings close to those of the clean encoder for selected samples, while still introducing the backdoor. This models an attacker who tries to **minimize embedding shifts to evade detection**.
>
> Even in this adaptive setting, fingerprint twins remain effective. The table below reports fingerprint-twin consistency rates under different encoders for a SimCLR-based CIFAR-10 encoder:
>
> | Downstream Task | Untampered Encoder | BadEncoder w/o adv loss Consistency Rate | BadEncoder w adv loss Consistency Rate |
> | --------------- | ------------------ | :--------------------------------------: | :------------------------------------: |
> | CIFAR‑10        | 98.00%             |                  25.00%                  |               **31.10%**               |
> | STL‑10          | 96.70%             |                  25.40%                  |               **32.30%**               |
> | GTSRB           | 84.10%             |                  18.70%                  |               **25.60%**               |
> | SVHN            | 83.80%             |                  24.80%                  |               **32.10%**               |
>
> While the adversarial loss slightly increases the consistency rate compared to the non-adaptive BadEncoder variant, **consistency remains dramatically lower than for the untampered encoder**. Standard hypothesis testing on these rates continues to reliably flag tampering.
>
> Thus, our fingerprint-twin method remains robust even against an adaptive attacker who explicitly tries to preserve embedding similarity.
>
> References
>
> [c] "Sensitive-sample fingerprinting of deep neural networks." CVPR, 2019.
>
> [d] "Towards stricter black-box integrity verification of deep neural network models." ACM MM, 2024.
>
> [e] "Intersecting-boundary-sensitive fingerprinting for tampering detection of DNN models." ICML, 2024.
>
> [h] "SDBF: Steep-Decision-Boundary Fingerprinting for Hard-Label Tampering Detection of DNN Models." CVPR, 2025.
>
> [i] "Badencoder: Backdoor attacks to pre-trained encoders in self-supervised learning." IEEE S&P, 2022.
>
> [j] BadEncoder official implementation: https://github.com/jjy1994/BadEncoder
>
> [k] "Distribution preserving backdoor attack in self-supervised learning." IEEE S&P, 2024.
>
> [l] DRUPE official implementation: https://github.com/Gwinhen/DRUPE
>
> [m] Optimum-Quanto quantization backend: https://github.com/huggingface/optimum-quanto
>
> [n] "Publiccheck: Public integrity verification for services of run-time deep models." IEEE S&P, 2023.

---

### Official Review · Reviewer_73vo · 2025-10-31

**Soundness:** 3
**Presentation:** 3
**Contribution:** 2
**Rating:** 4
**Confidence:** 3

**Summary:**

This paper addresses the integrity verification challenge of pre-trained encoders in the Encoder-as-a-Service (EaaS) paradigm under strict black-box settings—where only application outputs are observable, and internal model access or downstream pipeline knowledge is unavailable. Existing fingerprinting methods fail here as they rely on model predictions or task-specific information.
For any Lipschitz-continuous downstream function, larger divergence in encoder embeddings increases the likelihood of observable differences in application outputs. Building on this, the authors propose fingerprint twins—paired inputs optimized to produce nearly identical embeddings on intact encoders but sharply divergent embeddings after tampering. To simulate realistic tampering (e.g., backdoor injection, INT8 quantization), they design importance-aware perturbations that target low-importance weights (estimated via layer-wise normalized gradients under contrastive loss), ensuring tampered encoders retain normal performance while inducing detectable embedding shifts.

**Strengths:**

1-It addresses a practical, under-solved challenge in EaaS—black-box encoder integrity—where existing methods (relying on model predictions) are inapplicable. This aligns with real-world deployment needs (e.g., cloud-based encoder services) and avoids "solution looking for a problem" pitfalls.

2-Proposition 1 (linking embedding divergence to downstream output differences via Lipschitz continuity) provides a rigorous basis for fingerprint twin design, distinguishing the work from heuristic-based methods. The mathematical proof ensures the approach’s generality across downstream pipelines.

3-The combination of fingerprint twins and importance-aware perturbations is novel: Fingerprint twins solve the "downstream-agnostic" problem by focusing on embedding consistency, not task-specific outputs; Importance-aware perturbations mimic real attacker behavior (preserving performance while tampering), ensuring experiments reflect realistic threats.

**Weaknesses:**

1-The paper only evaluates vision encoders (ResNet-18 pre-trained via SimCLR/ImageNet). EaaS includes language (e.g., BERT) and multimodal (e.g., CLIP) encoders, which have distinct embedding spaces and tampering vectors (e.g., token-level backdoors in language models). Without validation on non-vision encoders, the method’s generalizability to EaaS as a whole is unproven.

2-Experiments only test static tampering (e.g., one-time backdoor injection, quantization). In practice, encoders may undergo dynamic, incremental changes (e.g., continuous fine-tuning for domain adaptation). The paper does not evaluate whether fingerprint twins retain detection ability over time or if retraining is required, limiting its applicability to long-running EaaS deployments.

3-For large-scale EaaS (e.g., cloud services with 1000+ encoders), the method’s fingerprint generation cost is unclear. The paper mentions generating 1000 twin pairs per encoder, but does not report time/compute overhead. Without scalability data, it is unknown if the method is feasible for large deployments.

**Questions:**

Please see Weaknesses.

---

> ### Author Response · Authors · 2025-12-03
>
> We thank the reviewer for the thorough reading and constructive feedback. Below we address each of the concerns and clarify how our method extends beyond vision, handles dynamic tampering, and remains practical in terms of fingerprint generation cost.
>
> ---
>
> ### W1 – Lack of evaluation beyond vision encoders
> To demonstrate that our method is not limited to vision encoders, we performed additional experiments on a non-vision encoder for different modailities, speech-to-text (ASR).
>
> We use a standard and popular pretrained audio encoder, Wav2Vec 2.0-Base, and evaluate it on an ASR downstream task. For consistency, we define agreement between fingerprint twins in terms of **word-error rate (WER)** between their transcribed outputs:
> - For each audio fingerprint-twin pair, we run both through the encoder and ASR stack and produce two transcripts.
> - We then compute the WER between the two transcripts.
> - A pair is counted as consistent (untampered) if its WER is below a fixed threshold (0.20 in our evaluation to keep low false-alarm while maintain high detection rate).
>
> We compare the clean encoder to a tampered encoder via INT8 quantization. The experimental setup for tampering detection on the speech encoder (Wav2Vec-Base–Librispeech) is:
>
> - **Encoder**: pretrained Wav2Vec‑Base speech encoder.
> - **Fingerprint generation**: 200 audio twin‑pairs on the untampered encoder.
> - **ASR downstream**: linear projection + CTC decoding to generate transcripts.
> - **Consistent criterion**: word‑error rate (WER) between twin outputs ＜ 0.20.
>
> The results are
>
> | Condition                           | Untampered↑   | Tampered↓         |
> | ----------------------------------- | ------------- | ------------------ |
> | Twin‑pair consistency rate          | **77.5%**     | **39.5%**          |
> | mean WER - clean samples            | 3.95%         | 4.32%              |
> | mean WER - FP twins                 | 12.50%        | 30.46%             |
>
> From the results, we can see that
> - The ASR performance on clean inputs remains essentially unchanged after tampering (mean WER 3.95% vs. 4.32%).
> - In contrast, twin-pair consistency drops significantly from 77.5% to 39.5%, and the mean WER between twin outputs more than doubles (12.50% → 30.46%).
>
> These ASR experiments provide strong evidence that fingerprint twins **generalize beyond vision**.The core mechanism, embedding instability leading to divergence in downstream outputs, extends naturally from images to speech encoders, under a realistic encoder-as-a-service threat model.
>
> ---
>
> ### W2 – Lack of evaluation beyond dynamic tampering
> The reviewer raised concerns that our method might be tailored to one-off tampering events and not to more gradual or dynamic changes.
> To address this, we conducted a controlled fine-tuning experiment on a self-supervised encoder and evaluated the consistency of fingerprint twins across multiple downstream tasks.
>
> We train an encoder from scratch using MoCov3 with a ResNet-18 backbone on CIFAR-10 for 1000 epochs. After ~500 epochs the encoder has **converged**. We select the checkpoint at epoch 600 as the untampered encoder and generate fingerprint-twin pairs on this model.
> Starting from this model, we further fine-tune the encoder for additional epochs, producing new checkpoints. For each checkpoint, we evaluate fingerprint-twin consistency rates (%) on several downstream classification tasks (CIFAR-10, STL-10, SVHN, GTSRB).
>
> |      Downstream\Training Epoch       | Untampered-600↑ |  Tampered-700↓  |  Tampered-800↓  |  Tampered-900↓  |  Tampered-1000↓  |
> | :------------------------------------: | :------------: | :---: | :---: | :---: | :---: |
> |   CIFAR‑10    |    95.4   | 45.0  | 52.7  | 47.2  | 47.0  |
> |    STL‑10     |    **95.7**    | 55.4  | 57.6  | 58.1  | 56.9  |
> |    SVHN       |    **92.4**    | 61.9  | 63.2  | 59.6  | 63.9  |
> |    GTSRB      |    **89.1**    | 46.1  | 53.5  | 49.2  | 47.2  |
>
> In these experiments, encoder and downstream performance remain nearly unchanged across checkpoints (accuracies vary < 1%), while consistency rates drop dramatically for the fine-tuned encoders.
> This demonstrates that our method is not restricted to detecting a single “before/after” tampering event. It **remains effective in dynamic tampering scenarios, where the encoder is gradually modified over time while its downstream performance appears stable.**

---

> > ### Author Response · Authors · 2025-12-03
> >
> > ### W3 – Method’s fingerprint generation cost
> >
> > As also discussed in our response to Reviewer kcRE (W3), fingerprint generation is a **one-time, offline operation** per encoder, performed by the model owner prior to deployment. The same fingerprint set can then be reused across many verification episodes and downstream tasks.
> >
> > We measured the **average time cost** to generate a single fingerprint-twin pair on a single NVIDIA A100 GPU in two representative settings:
> >
> > - Small model (CIFAR-10 + ResNet-18): **0.41 seconds** per fingerprint-twin pair.
> > - Large model (ImageNet + DINOv2-ViT-Base): **~40 seconds** per fingerprint-twin pair.
> >
> > In practice, the total cost is small relative to the pretraining or fine-tuning cost of such encoders. Fingerprint generation is parallel and can be distributed across GPUs. Once generated, fingerprints can be reused across multiple downstream applications and repeated integrity checks.
> > Thus, the overhead is modest even for large foundation-model-scale encoders, and is practical for large-scale encoder-as-a-service deployments.
> >
> > ---
> >
> > We hope these clarifications address the reviewer’s concerns and help convey that our method is **modality-agnostic**, **robust to dynamic tampering**, and **practically deployable** in real-world encoder-as-a-service scenarios.

---

### Official Review · Reviewer_kcRE · 2025-11-03

**Soundness:** 3
**Presentation:** 2
**Contribution:** 3
**Rating:** 6
**Confidence:** 4

**Summary:**

This paper tackles the problem of detecting tampering in pre-trained encoders deployed in black-box settings, where only downstream task outputs are accessible.
The authors propose fingerprint twins, paired inputs that produce nearly identical embeddings on an intact encoder but diverge after tampering. The method simulates realistic model manipulations via importance-aware weight perturbations and optimizes twins to maximize KL divergence while maintaining visual fidelity.
Extensive experiments across datasets, architectures, and attack types demonstrate that the proposed fingerprints can reliably distinguish tampered from untampered encoders with only a small number of queries, achieving strong transferability across downstream tasks and minimal impact on baseline utility .

**Strengths:**

1. Clear motivation and problem setup for strict black-box encoder integrity verification, covering a realistic Encoder-as-a-Service scenario.

2. Novel twin-based fingerprinting mechanism with theoretical connection to Lipschitz downstream transformations.

3. Strong empirical results across datasets (CIFAR-10/100, STL-10, GTSRB, SVHN) and tampering types (backdoor, DRUPE, quantization), including ablations exploring perturbation magnitude and subtle tampering.

4. Well-designed hypothesis-testing framework enabling low query complexity and downstream-agnostic evaluation.

**Weaknesses:**

1. Clarity of consistency-rate computation.
The paper heavily relies on the “consistency rate” metric for verification, but the operational definition and exact decision procedure could be more transparent. For instance, how consistency-rate is computed under arbitrary downstream tasks could be explained more formally.

2. Scalability concerns.
Although the method is tested on multiple datasets and even includes ImageNet-pretrained models, scalability to large-scale or higher-resolution encoders (e.g., ViT, 512×512 vision models) is unclear.

3. Overhead concerns.
The approach involves white-box access and iterative optimization for fingerprint generation; computational cost may become significant for foundation-model-scale encoders and there seems no discussion on this.

4. Perturbation simulation choices.
Manipulation simulation currently relies primarily on importance-aware weight perturbations.
It would be helpful to see analysis of more subtle real-world tampering, e.g., small-scale fine-tuning, channel-wise scaling, or feature-space transformations.
It is unclear how sensitive the method is to the specific perturbation design, and whether alternative (or more subtle) tampering regimes would produce high-quality twins with similar discriminative power.

5. Limited discussion of encoder-level defenses.
The paper would benefit from a deeper positioning against prior work on encoder-level security and robustness.
Several recent studies have the integrity or robustness of pretrained encoders[1][2][3].
Acknowledging these efforts and clarifying how this method differs, particularly in terms of threat model, deployment setting, and guarantees, would strengthen the narrative and highlight the contribution of fingerprint twins for black-box integrity verification.

[1] Zheng et al, SSL-cleanse: Trojan detection and mitigation in self-supervised learning. ECCV'24

[2] Feng et al., Detecting Backdoors in Pre-trained Encoders. CVPR'23

[3] Bansal et al., CleanCLIP: Mitigating Data Poisoning Attacks in Multimodal Contrastive Learning. ICCV'23

**Questions:**

Please respond to Weaknesses.

---

> ### Author Response · Authors · 2025-12-03
>
> We thank the reviewer for the detailed and constructive feedback. Below we address each concern in turn, and we will incorporate the corresponding clarifications and additional results into the revised manuscript.
>
> ---
> ### W1 – Clarity of consistency-rate computation
>
> As stated in Sec. 4.1 _(Experimental Setup – Evaluation Metrics)_, the consistency rate is defined as the fraction of fingerprint-twin pairs for which the downstream predictions **agree**. Concretely:
> - Classification downstream tasks.
> For each fingerprint-twin pair, we pass both inputs through the encoder and downstream classifier and compare their predicted class labels. If the **two labels are identical**, that pair is counted as consistent, and we regard the encoder as “untampered” for that query.
> -	Non-classification downstream tasks.
> For each fingerprint-twin pair, we measure similarity between the two outputs and treat the encoder as “untampered” for that query if the **similarity exceeds a predefined threshold**. For example, in speech-to-text, we apply the two twins, compare the resulting transcripts, and count the pair as consistent if the word-error rate (WER) between the two transcriptions is below a pre-specified threshold.
>
> We will clarify this definition and include these concrete type-specific explanations in the revised manuscript to avoid any ambiguity.
>
> ---
> ### W2 – Scalability to large-scale or high-resolution encoders
>
> To directly address this, we evaluated our framework on **a large, state-of-the-art vision encoder**, a DINOv2 Vision Transformer (ViT-Base) pretrained on large-scale data, with **ImageNet** as the downstream classification task.
> We then applied a subtle yet realistic tampering operation, INT8 post-training quantization (compression), which is representative of the kind of parameter-level change a service provider might apply to reduce cost while attempting to preserve accuracy. The results are:
>
> | Model / Condition           | Downstream Clean Accuracy            | Consistency Rate |
> | --------------------------- | ------------------------------------ | ---------------- |
> | Original model(untampered)  | 83.4%                              | 97.9%            |
> | After compression tampering | 83.3%                              | 71.3%            |
>
> Key observations include
> - **Downstream utility is preserved.** Accuracy changes by only 0.1% (83.4% → 83.3%), which would be extremely difficult to detect using accuracy-based monitoring alone.
> - **Consistency rate drops significantly.** The consistency rate falls from 97.9% to 71.3%, a drop of roughly 27 percentage points, despite the almost unchanged downstream accuracy.
> - **Sample-efficient detection.** Under our binomial testing framework (Sec. 3.5), this separation allows us to conduct tampering detection with both type-I and type-II error below 5% using only 15 fingerprint-twin probes.
>
> This experiment shows that our method (i) scales to modern, large-scale encoders and higher-resolution inputs, and (ii) remains highly sensitive to subtle, parameter-level tampering such as compression that barely affects task performance. We will integrate this result into the main experimental section and explicitly highlight it as evidence of scalability.
>
> ---
> ### W3 – Computational overhead of white-box fingerprint generation
> Fingerprint generation is a **one-time, offline process** performed by the model owner before deployment. We measured the average time cost for generating a twin pair on a single NVIDIA A100 GPU for two representative settings:
> - Small model (CIFAR-10 + ResNet-18): **0.41 seconds** per fingerprint-twin pair.
> - Large model (ImageNet + DINOv2-ViT-Base): **~40 seconds** per fingerprint-twin pair.
>
> In typical usage, the model owner would generate on the order of hundreds to a thousand twins once, and then reuse the same fingerprint set for many verification episodes and many downstream applications. Given that pretraining/fine-tuning these encoders already costs orders of magnitude more GPU time, and generation is fully parallelizable across GPUs and fingerprint pairs, we believe this overhead is modest for foundation-model-scale encoders and does not limit practicality.

---

> > ### Author Response · Authors · 2025-12-03
> >
> > ### W4 – Perturbation simulation choices and sensitivity to subtle changes
> >
> > To demonstrate sensitivity to small but realistic weight updates, we conducted a controlled **fine-tuning** experiment. We trained an encoder from scratch using self‑supervised learning (MoCov3+ResNet‑18) on CIFAR‑10, selected a converged checkpoint as the untampered model, and then performed **additional fine‑tuning** around converged point to simulate subtle parameter changes. The resulting consistency rates (%) are
> >
> > |   Dataset     | Untampered |  Fine-tuned  |
> > | :-----------: | :--------: | :---:        |
> > |   CIFAR‑10    |    95.4    | 45.0         |
> > |   STL‑10      |    95.7    | 55.4         |
> > |   SVHN        |    92.4    | 61.9         |
> > |   GTSRB       |    89.1    | 46.1         |
> >
> > Across multiple downstream tasks (CIFAR‑10, STL‑10, SVHN, GTSRB), encoder performance remained essentially unchanged (downstream accuracy variation < 1%), yet **the consistency rate of fingerprint twins drops sharply**, from approximately 89.1–95.7% (untampered) to 45.0–61.9% (fine-tuned). This clearly shows that **fingerprint twins amplify subtle weight-level modifications into large, easily detectable discrepancies**.
> >
> > Our threat model is explicitly focused on _parameter-level manipulations of the encoder_ (e.g., fine-tuning, backdoor injection, compression, quantization), which are the natural mechanisms by which a provider might alter a model while preserving the same interface.
> > In contrast, **feature-space transformations such as channel-wise scaling** applied after the encoder (i.e., transformations that do not modify encoder weights) are conceptually different. They alter the downstream pipeline rather than the pretrained encoder itself. Such operations do not constitute tampering with the certified encoder weights and are therefore outside the scope of the threat model we target (Sec. 3.1). We will clarify this explicitly in the revision to avoid confusion.
> >
> > In summary, the results above show that our method is sensitive to very modest fine-tuning that keeps utility almost unchanged, and that it is specifically designed for weight-level modifications, which is precisely the integrity-verification problem we aim to address.
> >
> > ---
> > ### W5 - Discussion of encoder-level defenses and relation to robustness
> >
> > We believe this concern stems from conflating two distinct notions:
> >
> > | Aspect                 | Robustness of Pretraining / Encoder Training                                                                                                    | Post-deployment Integrity Verification (our focus)                                                                                                                                                    |
> > | ---------------------- | ----------------------------------------------------------------------------------------------------------------------------------------------- | ----------------------------------------------------------------------------------------------------------------------------------------------------------------------------------------------------- |
> > | **Objective**          | Ensure the encoder is trained to be robust to adversarial or noisy data (e.g., robustness to poisoning or backdoor attacks during pretraining). | Determine whether a **deployed, pretrained encoder** has been **modified after certification** without the owner’s consent (e.g., via compression, quantization, fine-tuning, or backdoor insertion). |
> > | **Threat model**       | Attacks occur **at training time**, by manipulating the training data or training procedure.                                                    | Attacks occur **after deployment**, in a strict black-box setting where the encoder is embedded in unknown pipelines and only application outputs are observable.                                     |
> > | **Defenses / Methods** | Robust pretraining, robust loss functions, data sanitization, and related training-time defenses.                                               | Our method uses **fingerprint twins** and **downstream-agnostic hypothesis testing** to detect any **parameter-level change** to the encoder, regardless of how it was originally trained.            |
> >
> >
> > Thus, robust training and integrity verification are **complementary but fundamentally different**:
> > - Robust training aims to prevent certain attacks during pretraining.
> > - Our framework aims to **detect any post-deployment parameter modification of the encoder**, even if the original training procedure was robust.
> >
> > In the revised version, we will clarify that our contribution lies in the latter: black-box, post-deployment integrity checks for pretrained encoders embedded in opaque pipelines.
> >
> > ---
> >
> > We hope these clarifications address the reviewer’s concerns and better convey that our framework provides a **practical, scalable, and downstream-agnostic** integrity-verification tool for pretrained encoders under strict black-box constraints.

---

### Meta-Review · Area_Chair_L5WD · 2026-01-07

**Summary:**

The reviewers raised several concerns about this paper on fingerprint-based tampering detection for pre-trained encoders. Key concerns included scalability to large modern encoders, generalization beyond vision to other modalities, sensitivity to subtle tampering such as fine-tuning, and computational overhead. The discussion focused on whether the threat model is realistic, specifically the assumption of white-box access for fingerprint generation but only black-box access for verification, and how to distinguish tampering from legitimate encoder updates. The authors addressed most technical points via additional experiments on a large vision encoder and a speech model, plus results on fine-tuning and dynamic changes. However, deployment realism and the legitimate-update boundary remained partially unresolved for at least one reviewer, and presentation issues persist. Additionally, no reviewer strongly championed the paper.

**Reviewer Concerns:**

The authors provided a good rebuttal that addressed most technical concerns. They added experiments on large-scale vision encoders and speech encoders, showing the method generalizes across modalities and scales. They demonstrated detection of subtle fine-tuning and dynamic tampering scenarios. Computational overhead was clarified as a one-time offline cost that is reasonable relative to model training. New adaptive attacker experiments showed robustness against adversaries who try to preserve embedding similarity.

Howeverm reviewer qS2m's core objection about threat model applicability was not fully resolved, and questions persist about when encoder modifications are legitimate versus when they constitute tampering. The presentation also needs improvement, particularly around the consistency rate computation and positioning against related work. Given that one reviewer remains strongly negative and the threat model concerns are central to the paper's practical relevance, the paper would benefit from another revision cycle.

**Reviewer Scores:**

Reviewer kcRE started at 6 and would likely stay there or slightly increase given the added scalability and overhead clarifications. Reviewer 73vo started at 4 and would likely move into the accept range given the new non-vision and dynamic tampering results. Reviewer qS2m started at 2 and might increase modestly with the added large-scale evidence and clarifications, but would likely remain below the accept threshold due to skepticism about the threat model and deployment assumptions.

---

### Decision · Program_Chairs · 2026-01-26

Reject